# Proposal-Pokémon Battle Agent based on LLMs

Zihan Lv[1,*]     Qihang Cen[1,*]

2024316064, 2024210874
[1]Tsinghua University
{lvzh24, cqh24}@mails.tsinghua.edu.cn
[*]Equal contribution

## Abstract

The rapid development of LLMs has led to widespread applications in interactive environments, particularly in gaming, where LLM agents demonstrate impressive decision-making and strategy execution capabilities. This paper focuses on developing a Pokémon battle agent based on LLMs. We combine techniques such as supervised fine-tuning, KAG and Self-Consistency to improve agent's contextual understanding and generate effective battle commands. Through experiments on Pokémon Showdown with robots and human, we will evaluate the model's win rates and strategic performance, aiming to contribute to the development of LLM agents capable of dynamic and complex environments.

## 1   Background

The increasing popularity of interactive AI systems underscores the importance of developing engaging applications in gaming and education. Pokémon, as a well-loved franchise, presents a unique opportunity for natural language processing (NLP) systems to enhance user experience through dynamic interactions. Our project aims to address the challenge of creating a contextually aware Pokémon battle agent based on LLMs and explore the effectiveness of techniques such as supervised fine-tuning, knowledge-augmented generation, and self-consistency for outputting Pokémon battle commands from LLMs.

### 1.1   Pokémon

**Species:** Pokémon are diverse creatures that inhabit the world, each belonging to specific species that define their characteristics and abilities. For example, *Pikachu* is known for its electric abilities, while *Charizard* is recognized for its fire-based attacks.

**Type:** Each Pokémon belongs to one or more types, such as Fire, Water, or Grass, influencing their strengths and weaknesses. For instance, Water-types are strong against Fire-types but weak against Electric-types.

**Stats:** Pokémon have distinct stats that determine their battle performance, including HP, Attack, Defense, Special Attack, Special Defense, and Speed.

**Ability:** Every Pokémon possesses abilities that offer passive benefits during battles. For example, the ability of *Blaziken* is *Speed Boost*, which increases its Speed each turn, enhancing its offensive potential.

**Moves:** Pokémon can learn various moves that they can use in battles. These moves are categorized into different types and can have varying effects, including damage, status conditions, or healing. For

example, *Lucario* can learn *Aura Sphere*, a powerful Fighting-type move that never misses, while *Jigglypuff* might use *Sing* to put opponents to sleep. Trainers can strategize by selecting moves that best complement their Pokémon's types and abilities, further enhancing their effectiveness in battles.

## 2 Related Work

Generative AI and Large Language Models (LLMs) have demonstrated remarkable success in NLP tasks. A key future advancement will involve exploring how LLMs can autonomously operate in the physical world, extending their capabilities from text to action, which is crucial in the quest for Artificial General Intelligence. Games provide ideal environments for developing LLM-based agents that interact with virtual settings in ways that mimic human behavior. For instance, in communicative games like Werewolf [5], Avalane [2], agent could demonstrate strategic behaviors like human-beings. In sandbox games such as Minecraft, decision-making agents are designed to explore the world, acquire new skills, solve tasks, and create tools [4]. In tactical games like StarCraft2, agents learn strategic plannings and decisions, exhibiting comparable performance against the built-in AI and human players [3].

## 3 Proposed Method

The proposed method combines supervised fine-tuning of an LLM on a curated Pokémon dialogue dataset, incorporating knowledge-augmented generation to infuse domain-specific information and enhance contextual relevance. We plan to conduct experiments on the Pokémon Showdown platform[1], which is a web-based GUI for human players, and explore two technical approaches: (1) improving the knowledge-augmented generation and self-consistency approach in [1], and (2) improving the performance of LLMs through supervised fine-tuning. Dataset will include replays of battles on the platform. Referring to the method in [1], we evaluate the ability of large language models by comparing their win rates and other indicators.

In approach 1, we leverage KAG to integrate external knowledge sources such as Pokémon type advantages, move effects, and ability interactions. By incorporating domain-specific information, the model will reduce hallucinations fundamentally. Additionally, techniques such as Chain-of-Thought (CoT) and Self-Consistency are used to improve complex reasoning. These methods work together to boost the model's decision-making capability in dynamic and complex battle scenarios.

In approch 2, we use structured battle data from Pokémon Showdown replays to fine-tune the LLM, allowing it to learn from examples of successful strategies and decision-making processes. By providing the model with high-quality, formatted battle sequences, it improves the LLM's ability to understand and adapt to game strategies. The supervised fine-tuning focuses on improving the model's comprehension of in-game dynamics, increasing its tactical accuracy and adaptability in Pokémon battles.

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
