# OpenReview forum: "【Proposal】Pokémon Battle Agent based on LLMs"
_tsinghua.edu.cn/THU/2024/Fall/AML — THU 2024 Fall AML Submission_

### Official Review · ~Zijun_Liu2 · 2024-11-08
**Review and Feedback**

**Rating:** 8
**Confidence:** 3

**Review:**

## Overview
The proposal for a Pokémon battle agent leveraging large language models (LLMs) is intriguing, targeting the development of an AI agent that can effectively strategize within Pokémon battles. The authors aim to achieve this by utilizing supervised fine-tuning, knowledge-augmented generation (KAG), and self-consistency to enhance contextual understanding and tactical decision-making. The project’s experiment design is clear to evaluate win rates and strategic capability against both automated systems and human opponents on Pokémon Showdown.

## Strengths
1. **Clear Focus on Techniques**: The proposal systematically addresses key techniques—KAG, Chain-of-Thought (CoT), and self-consistency. Each method has a distinct purpose, showing a thoughtful approach to the technical challenges of developing a decision-making agent.

2. **Relevance to Broader Research Trends**: By positioning this project within the context of games as testing environments for general AI, the proposal aligns with trends in AI research exploring LLMs in strategic, interactive scenarios. This adds value by contributing to foundational research that might inform the development of autonomous decision-making agents in varied applications beyond gaming.

## Suggestions for Improvement
1. **More Reference Needed**: The proposal could benefit from more references on some terms mentioned, e.g., KAG. This will help readers understand the context and relevance of these techniques in the broader AI literature.

2. **Novelty Statement**: The proposal could be strengthened by a more explicit statement on the novelty of the project. Currently, approach 1 shows bare differences from existing methods. Highlighting the unique aspects of the project would make it more compelling. Still, the novelty of the project is clear in approach 2.

---

### Official Review · ~Kangping_Xu1 · 2024-11-09
**Review of "Pokémon Battle Agent based on LLMs"**

**Rating:** 8
**Confidence:** 4

**Review:**

## Pros
- **Innovative Approach**: Leveraging LLMs to develop an interactive game agent demonstrates the potential of AI beyond conventional text-based tasks, particularly in dynamic and strategic environments like Pokémon battles.
- **Comprehensive Techniques**: The integration of KAG and Chain-of-Thought reasoning provides a strong foundation for improving contextual accuracy and decision-making, reducing model hallucinations.

## Cons
- **Dataset Limitations**: Relying solely on Pokémon Showdown battle data may introduce biases, limiting the model's adaptability to novel scenarios outside the dataset's scope, so I think online data collection and RL methods may help.

Overall, this project holds promise for advancing AI capabilities in gaming, but its success will depend on overcoming challenges related to data diversity and computational resource demands. By improving LLMs' contextual understanding in dynamic environments, this project could pave the way for more sophisticated AI-driven game interactions.

---

### Official Review · ~Lei_Wu17 · 2024-11-09
**Evaluation of "Pokémon Battle Agent based on LLMs" Proposal**

**Rating:** 7
**Confidence:** 4

**Review:**

# Pros
* Well-Defined Scope: The proposal is focused, tackling specific AI techniques in a particular gaming context, which should enable meaningful experimental results.
* Innovative Use of LLMs: Utilizing LLMs in a tactical and strategic context is a novel approach that could reveal insights into AI-based decision-making.
* Detailed Methodology: The proposal outlines specific techniques such as supervised fine-tuning, KAG, and CoT, indicating a structured approach to improve the model’s capabilities.
* Clear Evaluation Metrics: The plan to measure win rates and strategic performance on Pokémon Showdown provides tangible, quantifiable metrics for evaluating the model.
# Cons
* Lack of Specificity in Implementation: The proposal could benefit from more detail on data handling and preprocessing, or how KAG and CoT will be implemented in practice.
* Limited Discussion on Limitations: There is little mention of potential limitations or challenges, such as computational resource demands or handling unpredictable human strategies.
* Risk of Overfitting to a Niche Context: Fine-tuning for Pokémon battles may yield results that are not easily generalizable to other interactive settings, limiting broader applicability.
* Complexity of Evaluation: Evaluating LLMs in dynamic games may pose challenges, particularly in assessing "strategic performance" without a clear set of benchmarks.

---

### Official Review · ~Bryan_Constantine_Sadihin1 · 2024-11-09
**Review of "Pokemon Battle Agent based on LLMs"**

**Rating:** 8
**Confidence:** 4

**Review:**

Strengths:
1. Innovative Application: The topic of using LLM within Pokemon battle context is a creative approach, which showcases the potential of LLMs in interactive gaming environments.
2. Specific defined Techniques: The proposal has defined some probable method that is going to be used, such as KAG and Self-Consistency, alongside fine-tuning with battle replays data.

Cons:
1. Inexistence of Metric: The paper could have been improved by mentioning specific optimization metric for training, validation, and testing
2. Limited Generalizability: As the focus of this paper is solely for Pokemon battles, it limits the direct applicability beyond this domain. Applying to other domain can't directly implementing the same research without major modification.

---

### Official Review · ~Chentian_wei1 · 2024-11-10
**This paper intriguingly integrates popular LLM methods for Pokémon battles, but the method's breadth and lack of definition could benefit from a clearer framework description.**

**Rating:** 7
**Confidence:** 4

**Review:**

The work presented in this paper is quite interesting, integrating several of the most popular LLM methods currently available and applying them to the specific task of Pokémon battles, with the definition and rules of the Pokémon game being very clear. However, the proposed method seems somewhat too broad and lacks clear definitions. Additionally, it might be beneficial to provide a slightly more detailed introduction and description of the overall framework.

---

### Official Review · ~Keyu_Shen1 · 2024-11-10
**Well-structured Proposal**

**Rating:** 7
**Confidence:** 3

**Review:**

The proposal offers an innovative application of large language models (LLMs) in the context of Pokémon battles, showcasing the potential of LLMs in strategic, interactive gaming environments. The project combines techniques like supervised fine-tuning, knowledge-augmented generation (KAG), and self-consistency to improve decision-making and context understanding. However, the proposal could benefit from more defined evaluation metrics and a clearer statement on its novelty.

---

### Official Review · ~Jin_Zhu_Xu1 · 2024-11-11
**Clear idea topic**

**Rating:** 7
**Confidence:** 4

**Review:**

The proposal explains a clear and creative idea topic, but not convincing enough on how the proposed techniques is applicable to the targeted objectives. The proposed techniques explanation is too general and not clear on how specific technical improvements will impact gameplay mechanics, as well as how the evaluation framework works

---

### Official Review · ~Ziyu_Zhao6 · 2024-11-11
**Review of "Pokémon Battle Agent based on LLMs" Proposal**

**Rating:** 7
**Confidence:** 4

**Review:**

Overview:
This proposal presents a project to develop a contextually aware Pokémon battle agent using Large Language Models (LLMs). The project aims to enhance user engagement by allowing the agent to output Pokémon battle commands dynamically and accurately. The proposed method combines two main approaches: supervised fine-tuning on Pokémon battle datasets and knowledge-augmented generation (KAG) with self-consistency mechanisms.

Strengths:
	1. Novel and Engaging Use Case: The proposal leverages a highly popular and relatable franchise (Pokémon) to explore the potential of LLMs in an interactive gaming environment, which could draw significant interest in gaming and education sectors.
	2. Integration of Advanced NLP Techniques: By combining knowledge-augmented generation and self-consistency techniques, the proposal addresses a key challenge in generative AI—reducing hallucinations and improving contextual understanding.

Weaknesses:
	1. Method Novelty: While KAG and self-consistency are proposed to mitigate hallucinations, the proposal's novel points are not well clearly stated compared to related works.
	2. Evaluation Metrics Ambiguity: Although win rates are mentioned, other performance metrics (e.g., decision-making latency and user satisfaction) are not discussed in detail. Adding these would provide a more comprehensive evaluation framework for the proposed system’s success in real-world applications.

---

### Official Review · ~Gangxin_Xu1 · 2024-11-12
**Review of "Pokémon Battle Agent based on LLMs"**

**Rating:** 8
**Confidence:** 4

**Review:**

This proposal introduces a large language model (LLM)-based agent designed for Pokémon battles, leveraging techniques like supervised fine-tuning (SFT), knowledge-augmented generation (KAG), and self-consistency. The goal is to create an agent with enhanced contextual understanding and strategic decision-making abilities. Evaluation will involve testing the agent's performance in Pokémon Showdown battles against both AI and human opponents, with a focus on win rates and strategic effectiveness.

Strengths:
Novel Application in Gaming: Developing an LLM agent specifically for Pokémon battles is an innovative application, as it combines complex decision-making, strategy, and adaptability in a well-defined environment.
Robust Methodology: The integration of SFT, KAG, and self-consistency reflects a well-rounded approach to enhancing the agent's contextual and strategic capabilities, addressing the nuances of battle decisions.
Clear Evaluation Plan: The proposal outlines a straightforward and relevant evaluation method through win rates and strategic assessment in Pokémon Showdown, providing concrete metrics to gauge the agent’s performance.

---

### Official Review · ~Kittaphot_Saengprachathanarak1 · 2024-11-12
**Review of "Pokemon Battle Agent based on LLMs"**

**Rating:** 8
**Confidence:** 4

**Review:**

The paper presents a Pokémon battle agent based on large language models (LLMs) that aims to improve agent performance through supervised fine-tuning, knowledge-augmented generation (KAG), and self-consistency. By incorporating domain-specific knowledge and enhancing the model’s reasoning, it attempts to build a contextually aware agent capable of handling dynamic battle scenarios. The experiments will be conducted using the Pokémon Showdown platform, where the agent's strategic decision-making will be evaluated against both robots and humans. While the proposed methods, such as integrating external knowledge and fine-tuning with battle replays, are innovative, further validation and comparison with existing Pokémon battle agents would strengthen the contribution. This work is an interesting step forward in using LLMs for dynamic, game-based decision-making.